# Novel QTL Associated with Aerenchyma-Mediated Radial Oxygen Loss (ROL) in Rice (*Oryza* *sativa* L.) under Iron (II) Sulfide

**DOI:** 10.3390/plants11060788

**Published:** 2022-03-16

**Authors:** Dang Van Duyen, Youngho Kwon, Nkulu Rolly Kabange, Ji-Yoon Lee, So-Myeong Lee, Ju-Won Kang, Hyeonjin Park, Jin-Kyung Cha, Jun-Hyeon Cho, Dongjin Shin, Jong-Hee Lee

**Affiliations:** 1Molecular Biology Department, Agricultural Genetic Institute, Hanoi 11917, Vietnam; dangvanduyen79@gmail.com; 2Department of Southern Area Crop Science, National Institute of Crop Science, RDA, Miryang 50424, Korea; kwon6344@korea.kr (Y.K.); minitia@korea.kr (J.-Y.L.); olivetti90@korea.kr (S.-M.L.); kangjw81@korea.kr (J.-W.K.); tinapark@korea.kr (H.P.); jknzz5@korea.kr (J.-K.C.); hy4779@korea.kr (J.-H.C.); jacob1223@korea.kr (D.S.)

**Keywords:** radial oxygen loss, aerenchyma, quantitative trait locus, greenhouse gas, rice

## Abstract

In rice, high radial oxygen loss (ROL) has been associated with the reduction in the activity of methanogens, therefore reducing the formation of methane (CH_4_) due to the abundance in application of nitrogen (N)-rich fertilizers. In this study, we evaluated the root growth behavior and ROL rate of a doubled haploid (DH) population (*n* = 117) and parental lines 93-11 (P1, *indica*) and Milyang352 (P2, *japonica*) in response to iron (II) sulfide (FeS). In addition, we performed a linkage mapping and quantitative trait locus (QTL) analysis on the same population for the target traits. The results of the phenotypic evaluation revealed that parental lines had distinctive root growth and ROL patterns, with 93-11 (*indica*) and Milyang352 (*japonica*) showing low and high ROL rates, respectively. This was also reflected in their derived population, indicating that 93.2% of the DH lines exhibited a high ROL rate and about 6.8% had a low ROL pattern. Furthermore, the QTL and linkage map analysis detected two QTLs associated with the control of ROL and root area on chromosomes 2 (*qROL-2-1*, 127 cM, logarithm of the odds (LOD) 3.04, phenotypic variation explained (PVE) 11.61%) and 8 (*qRA-8-1*, 97 cM, LOD 4.394, PVE 15.95%), respectively. The positive additive effect (2.532) of *qROL-2-1* indicates that the allele from 93-11 contributed to the observed phenotypic variation for ROL. The breakthrough is that the *qROL-2-1* harbors genes proposed to be involved in stress signaling, defense response mechanisms, and transcriptional regulation, among others. The qPCR results revealed that the majority of genes harbored by the *qROL-2-1* recorded a higher transcript accumulation level in Milyang352 over time compared to 93-11. Another set of genes exhibited a high transcript abundance in P1 compared to P2, while a few were differentially regulated between both parents. Therefore, *OsTCP7* and *OsMYB21*, *OsARF8* genes encoding transcription factors (TFs), coupled with *OsTRX*, *OsWBC8,* and *OsLRR2* are suggested to play important roles in the positive regulation of ROL in rice. However, the recorded differential expression of *OsDEF7* and *OsEXPA*, and the decrease in *OsNIP2*, *Oscb5*, and *OsPLIM2a* TF expression between parental lines proposes them as being involved in the control of oxygen flux level in rice roots.

## 1. Introduction

*Oryza sativa* L. remains the most cultivated species of rice across the world and comprises the *indica* and *japonica* subspecies [1]. The two rice subspecies are reported to have a large genetic variation [2,3], which is partially explained by their evolutionary genetic differentiation and their independent domestication histories [4]. Since its domestication, rice has remained, even today, as the only cereal crop mainly cultivated for human consumption, and half of the world’s population is said to depend on rice as a staple food, with a relatively small portion of global rice production and by-products used for animal feeding [5,6,7]. Rice cultivation requires an abundant use of nitrogen (N)-rich fertilizers from both organic and inorganic N sources, especially in high yielding varieties grown in lowland, irrigated, and flooded paddy fields [8,9]. However, despite the benefits of N in agriculture, excessive applications of N-rich fertilizers have been shown to contribute to the emission of greenhouse gases (GHGs), such as methane (CH_4_) and nitrous oxide (N_2_O) [10,11,12,13].

In the context of climate change, and regarding the necessity for reducing the emission of GHGs from agricultural activities, such as crop cultivation and livestock, there is a growing interest in the agricultural research community and plant breeding-related research in developing crop varieties with an increased nitrogen use efficiency (NUE), taking advantage of the panel of emerging modern breeding technologies and molecular breeding techniques [14,15,16]. The advent of genome-wide association studies (GWAS) [17] and the emergence of omics [18] as well as genome sequencing technologies [19,20] have revolutionized our understanding of the genetic factors underlying plants’ tolerance or resistance to environmental and biotic stresses, and offered a wide range of opportunities to develop and release, in a relatively short period of time, improved crop varieties for large scale production.

In plants, roots have a wide range of functions that contribute to their fitness and survival throughout their life cycle. In addition to serving as the structural support for the plant, roots mediate nutrient elements and water acquisition or uptake from the soil [21,22]. The rooting system of plants has a complex architecture; nonetheless, it is well-structured and functionally organized to the extent that each part of the rooting system plays a specific role, going from root growth from the root tip vertically to the root hairs or lateral roots horizontally. Root hairs, also known as absorbent hairs, are tubular outgrowths or a trichoblast, a hair-forming cell on the epidermis of a plant’s root after differentiation, that significantly increase the volume of soil that plants mine for nutrients [23]. Despite their role in absorbing nutrients and water as well as other minerals from the soil [24,25,26], root hairs also serve to release oxygen (O_2_) that is captured from the atmosphere by leaves and moves through the vessels down to the roots during respiration and photosynthesis [27,28,29,30].

The flow of O_2_ within the plant is facilitated by tissue composed of a network of interconnected gas conducting intercellular spaces called aerenchyma, which provide roots with O_2_ under hypoxic conditions [31]. This process is known to enhance the internal diffusion of atmospheric and photosynthetic O_2_ from the aerial part of the plant to the flooded roots, therefore allowing the roots to maintain aerobic respiration [32]. Studies have reported two types of aerenchyma; the cortical aerenchyma (primary aerenchyma) is formed in the roots of plant species such as rice, among other cereals crop species [33]. The formation of aerenchyma occurs in plants grown under hypoxic or flooded conditions, and may contribute to long-term avoidance of anoxia under flooded conditions, and improve crop growth and yield. Reports have indicated that there are two major types of primary aerenchyma identified, including schizogenous and lysigenous aerenchyma [34,35]. The first type develops gas spaces through the separation cell and differential cell expansion without cell death, while the second type is formed by the death (arises from spatially selective death of grown cells or a result of the programmed cell death) and subsequent lysis of some cells mainly in cereal crops, including rice [36]. Lysigenous aerenchyma can readily be promoted by soil waterlogging or partial shoot submergence and is said to be a prominent feature of several crop species, including rice.

The O_2_ that is released through the aerenchyma to the rhizosphere of the root system and the immediate environment is called radial oxygen loss (ROL) [37]. ROL helps reduce the accumulation of phytotoxins, such as sulfides and ferrous iron, via the rhizosphere oxidation [38]. Although it has been reported that the degree of CH_4_ from paddy fields occurs in a variety-dependent manner [10,39,40], high ROL has also been proposed to result in the reduction of CH_4_ [41]. Similarly, an increase in the porosity of adventitious roots of monocot wetland species grown under aerated conditions has been reported, and the authors suggested that ROL from the basal zones of roots would decrease longitudinal diffusion of O_2_ to the root apex; therefore, limiting the maximum penetration depth of these roots into anaerobic soil [42]. It is well established that large amounts of CH_4_ formation occurs under anaerobic conditions in flooded rhizospheres [43], which is also affected by, but not limited to, soil conditions and environmental factors such as the redox potential (Eh) and the hydrogen potential (pH) [43,44]. In the presence of O_2_, the activity of methanogens reduces significantly, which exerts an impact on the methanogenesis and the microbial community in the rhizosphere [45,46], but their sensitivity to O_2_ is said to vary among species [47,48].

Therefore, in the perspective of depicting the molecular basis and genetic factors influencing ROL in plants, we performed a linkage mapping and quantitative trait locus (QTL) analysis on a doubled haploid population grown on agar medium modified with iron (II) sulfide (FeS). Under the same conditions, the transcriptional regulation of a set of genes harbored by *qROL-2-1*, which their annotations suggested to be involved in stress response mechanisms, was investigated by qPCR.

## 2. Results

### 2.1. Growth Patterns of Doubled Haploid Lines in Response to FeS Treatment

FeS induces stress in plants, and the seedling stage is thought to be the most vulnerable point. Because our major target tissues were the roots, we evaluated the roots’ growth behavior of the mapping population along with their parental lines grown under FeS-modified medium conditions. The results indicate a normal distribution for ROL (Figure 1A) and root area (RA) (Figure 1B). However, a negative skewness was observed for root length (RL) under the same conditions (Figure 1C, Appendix A).

Initially, parental lines were evaluated for their phenotypic response towards FeS treatment. Results revealed that 93-11 (P1) had a low ROL rate of about 15.5% (Figure 2A,D), while Milyang352 recorded a high ROL rate of about 35.9%, which is about 2.3-fold compared to P1 (Figure 2A,E). However, P1 showed a high RA percentage (12.7%) compared to P2 (9.1%). In the same way, 93-11 had relatively shorter roots (6.8 cm) and Milyang352 had longer roots (8.9 cm). The recorded ROL, RA, and RL values for P1 and P2 were used as references to estimate the changes in their derived population. Our data indicate that nearly 93.2% of the mapping population exhibited a high ROL rate (P2-like, 16.5–49%), while about 6.8% had a low ROL rate (P1-like, ≤16.4%) (Figure 2A and Figure 3A). The lowest ROL rate recorded in the study was 12.1%, and the highest ROL observed was 49%. In addition, about 98.3% of the DH lines showed a relatively narrow RA (P2-like, ≤12.7%) (Figure 2B and Figure 3B), whereas about 1.7% recorded a wide RA (P1-like, >12.7%). Furthermore, about 43.6% and 56% of the mapping population had relatively short (P1-like, ≤6.8%) and long (P2-like, >6.8%) roots, respectively (Figure 2C and Figure 3C).

### 2.2. Principal Component Analysis (PCA) and Clustering of the Mapping Population

The DH lines (*n* = 117) were grouped into two distinct clusters based on their ROL phenotypic resemblance with either P1 or P2 (Figure 3A–C). In addition, ROL and RL showed similarity in their color schemes (increase or decrease patterns), and a fall in the same cluster as indicated by the dendrogram in panel D of Figure 3. We were also interested to investigate the existence of a possible relationship between ROL and other traits of interest in the mapping population. The results of the PCA (Figure 3E) suggest a weak correlation between RL and ROL. However, no correlation was found between root area and ROL. Principal components 1 (PC1) and 2 (PC2) explained 37.6% and 26.6% (for ROL), 35.6% and 28% (for RA), and 44.7% and 25% (for RL) of the proportion of variance of the observed phenotypes, respectively, of the mapping population grown under FeS conditions.

### 2.3. Quantitative Trait Loci (QTLs) Associated with Radial Oxygen Loss and Root Area under FeS in Rice

In this study, we used 105 Kompetitive Allele-Specific (KASP) and 135 Fluidigm markers, and the phenotype data of 117 DH rice lines and the parental lines, grown on a modified MS medium with FeS, to perform linkage mapping and QTL analysis. The results show that two QTLs, *qROL-2-1* (Chromosome 2, logarithm of the odds (LOD) 3.04, 127 cM) and *qRA-8-1* (LOD 4.39, 97 cM), associated with ROL and RA, respectively, were identified. The *qROL-2-1* is flanked by KJ02_047 (closest marker) and ad02011845 markers, while the *qRA-8-1* QTL is flanked by cmb0824_7 and GW8—AG (closest marker) (Table 1, Figure 4A,B). The additive effect observed for *qROL-2-1* (2.53) and that of *qRA-8-1* (0.76) indicate that the alleles from 93-11 contributed to the observed phenotypic variation for both ROL and RA under FeS treatment. The length of the linkage groups (LGs, with LG2 harboring *qROL-2-1* having a total length of 189.28 cM) and the density of KASP and Fluidigm markers across the rice genome is shown in Appendix A and Appendix A, respectively.

The *qROL-2-1* is the only QTL identified by the present study, which was associated with the control of ROL in rice. Therefore, we were interested to unveil the identity of genes harbored by this QTL region (745,184 bp, Chr2:24954729..25699913). Table 2 presents genes likely to be involved in the regulation (promoting or reducing) of ROL from the aerenchyma area of rice roots. The first category of the identified genes is proposed to have a DNA binding transcription factor (TF) activity as part of the genes’ specific molecular functions, and that they are involved in biological processes such as the response to endogenous stimuli, defense response, biosynthetic process, metabolic or cellular processes, and transcriptional regulation. Among them, we found *ARF8* TF (Os02g41800), an auxin responsive factor previously reported to play a key role in the auxin signaling pathway [49]. In the same way, *OsPLIM2a* TF (Os02g42820), belonging to the LIN-11, Isl1, and MEC-3 (LIM) domain-containing protein, having a double-zinc finger motif, was identified. Velyvis and Qin [50] supported that *OsPLIM2a* is involved in the development of rice seed and tillers. Another set of genes encoding TF found in the ROL QTL includes *OsMYB21* TF (myeloblastosis 21, Os02g42850), an AP2 domain-containing protein (Os02g42580), *OsTCP7* (Os02g42670), a member of the TEOSINTE-BRANCHED1/CYCLOIDEA/PCF7 (TCP7) TF family, and *OsDLN61* (Os02g42200), a DLN Repressor 61 and B-block binding subunit of TFIIIC domain-containing protein associated with transcription initiation from the RNA polymerase III promoter. We also identified *OsNIP2* or *OsLis1* (Os02g41860), a gene encoding aquaporin. In plants, aquaporin proteins are known for their roles in abiotic stress response [51,52] and the transport of multiple biological compounds, including water [53,54], arsenite [55], and silicon [56,57]. In the same group, a gene encoding the DUF (domain of unknown function, *OsDUF1771,* Os02g42670) is proposed to be involved in the regulation of epigenetic gene expression. Another gene of the same category, the *OsDEF7/OsCAL1/OsAFP1* (Os02g41904, a defensin-like DEFL family protein associated with the defense response mechanism), was identified as well.

Similarly, we could also find in the same locus, genes with interesting annotated functions, such as those associated with cellular signaling and signal transduction upon stress induction. These include the phytosulfokine receptor precursor (*OsPSKR1,* Os02g41860), the ATP binding cassette (ABC)-2 type transporter (*OsWBCH8,* Os02g41920), the A-TYPE response regulator receiver (*OsRRA8* or *OsRR11*, Os02g42060), the wall-associated receptor kinase 22 precursor (*OsWee1*, Os02g42110), *OsWAK14* (EGF-like calcium-binding domain containing protein, Os02g42150), phospholipase (lipase, class 3 domain-containing protein, Os02g42170), WD-40 repeat family protein (Os02g42590), double-stranded RNA-binding motif containing protein (Os02g42600), a thioredoxin (Os02g42700, having an enzyme regulator activity), and Os02g42790 encoding a short-chain dehydrogenase/reductase, having a catalytic activity.

Likewise, a number of genes involved in protein modification, the metabolic process or biosynthetic process, catalytic activity, oxidation–reduction (redox), or transport activity were detected as well. Of this number, we could find *OsWAK14* (Os02g42150), a wall-associated kinase also known as a receptor-like protein kinase or EGF-like calcium-binding domain-containing protein. In addition, a white-brown complex homolog protein 8 encoding gene (Os02g41920, harboring an ABC-2 type transporter domain-containing protein), a lectin receptor-type protein kinase (Os02g42780), AP2 domain containing protein (Os02g42585), a UDP-glucoronosyl/UDP-glucosyltransferase family protein encoding gene (Os02g42280) having a transferase activity, a ubiquitin-conjugating enzyme (E2) encoding gene (Os02g42314, harboring the RWD-like domain) having a ligase activity, a leucine-rich repeat 2 (Os02g42412, a cysteine-containing subtype protein) having an RNA-directed DNA polymerase activity, a dehydrogenase (Os02g42520) having a binding and catalytic activity, and a ferredoxin-thioredoxin reductase (Os02g42570, having a catalytic activity) associated with the photosynthesis were among the *qROL-2-1* QTL-related genes. To these genes, we can associate genes encoding an expansin precursor (Os02g42650, involved in anatomical structure morphogenesis), a degenerative spermatocyte homolog 1 (Os02g42660, encoding a migration-inducing gene 15 protein, having a sphingolipid delta-4 desaturase activity, an oxidoreductase activity, acting on paired donors, with the incorporation or reduction of molecular oxygen), a Zinc finger C3HC4 type domain-containing protein (Os02g42690) associated with the transport, protein modification, and catalytic activity, and a cytochrome b5-like heme (Os02g42740, harboring a steroid binding domain-containing protein), and an oxidoreductase (Os02g42810, belonging to the short-chain dehydrogenase/reductase family domain-containing protein, having a catalytic activity).

Moreover, the majority of genes harbored by the *qROL-2-1* QTL are localized in the membrane (cell membrane, cell wall, plasma membrane), but a few are shown to be localized in the nucleus, cytoplasm, and the plastid (Table 2).

### 2.4. Validation of qROL-2-1-Related Genes Associated with Radial Oxygen Loss in Rice

We were interested to see the changes in the transcript accumulation of a set of genes found in the *qROL-2-1* region, and having the potential to regulate ROL in rice roots based on their annotations and their predicted functional similarity with previously reported genes. The qPCR results indicate that, soon after FeS application, the transcript accumulation level of the *TEOSINTE-BRANCHED1CYCLODOIDEA/PCF7* (*OsTCP7*, Os02g42380) (Figure 5A), the thioredoxin encoding gene (*OsTRX*, Os02g42700) (Figure 5B), the myeloblastosis 21 (*OsMYB21*, Os02g42850), (Figure 5C), the *auxin responsive factor 8* (*OsARF8*, Os02g41800), and that of the white-brown complex homolog protein encoding gene *OsWBC8* (Os02g41920) increased over time upon FeS treatment, and was significantly induced at 6 h in the roots of Milyang352 (P2, high ROL cultivar) compared to 93-11 (P2, low ROL cultivar). In contrast, data in panels D, E, and I of Figure 5 reveal that the low ROL cultivar 93-11 recorded a sustained and significant increase in the transcript accumulation of the LIM domain protein encoding gene *OsPLIM2a* (Os02g42820), that of the defensin 7 encoding gene, also known as *OsCAL1* or *OsAFP1* (*OsDEF7*, Os02g41904), and the cytochrome b5-like heme/steroid binding domain encoding gene *Oscb5* (Os02g42740) compared to that observed in Milyang352, under the same conditions.

Unlike the genes described in the previous paragraph, the expression of *OsEXPA* (encoding an expansin, Os02g42650) (Figure 5G) and that of *OsNIP2*, also referred to as *OsLsi1* (encoding the aquaporin protein; Os02g41860) (Figure 5H), was significantly downregulated over time in the roots of Milyang352 (P2 with a high ROL rate), whereas, when expressed in 93-11 (P1 with a low ROL rate), *OsNIP2* exhibited a differential transcript accumulation pattern between 93-11 and Milyang352. Under the same conditions, the transcript accumulation of *OsEXPA* slightly decreased in 93-11 at 3 h of FeS application, but a significant difference was observed at 6 h. In the same way, the expression of a leucine-rich repeat 2 encoding gene (*OsLRR2*, Os02g42412) was significantly upregulated in the 93-11 background 3 h after FeS treatment but showed similar patterns in both 93-11 and Milyang352 at 6 h (Figure 5J). Of all analyzed *qROL-2-1*-associated genes in Milyang352, *OsTRX*, *OsMYB21* TF, and *OsWBC8* recorded the highest expression level at 6 h of FeS application (in descending order). However, *OsDEF7*, *OsNIP2*, and *OsPLIM2a* exhibited the highest expression level in the low ROL cultivar 93-11, after 6 h of FeS treatment.

In the perspective of providing insights into the basic mechanism underlying the response of plants to FeS-induced ROL in rice roots, we summarized the results and proposed a hypothetical signaling model primarily constructed using the recorded phenotypes of parental lines (93-11 and Milyang352 showing low and high ROL rates, respectively), coupled with their roots’ growth behaviors. To the phenotypes, we associated the recorded transcriptional response of the target genes harbored by the *qROL-2-1* in both parents under the same conditions (Figure 6). Genes for which the change in the transcriptional regulation patterns has been investigated are proposed to be involved in the adaptive response mechanisms towards abiotic or biotic stress based on their annotations or similarity with previously reported genes.

## 3. Discussion

### 3.1. Differential Radial Oxygen Loss between Parental Lines and Doubled Haploid Rice Lines

In East Asia, rice is often cultivated in flood-prone areas (paddy fields), wetland, or lowland characterized by hypoxic soil environments and chemically reduced because of the slow diffusion of O_2_ in water, and the rapid consumption of O_2_ by soil microorganisms [58,59]. Aeration in plants is crucial for the growth of roots under waterlogged conditions. It has been evidenced that some wetland plants form a structural barrier that impedes O_2_ leakage from the basal part of roots, identified as the ROL barrier, which reduces the loss of oxygen transporter via the aerenchyma to the tip of the root [60]. Plants subjected to waterlogged conditions rely on O_2_ supply from the aerial parts of the plant, whereas strong barriers to ROL, thick roots, and increased root porosity help to maintain a balanced supply of O_2_ to root apices [61,62,63]. A study measuring O_2_ permeability coefficients of rice roots [64] revealed that the resistance of the outer part of the root (OPR, a term referring to the four cell layers at the root periphery (rhizodermis, exodermis, sclerenchyma, and a layer of cortical cells, separated from the central cylinder by well-developed aerenchyma in the central cortex of rice roots)) to O_2_, rather than the axial diffusion via aerenchyma, was a rate-limiting factor. Here, we recorded a low ROL rate in 93-11 (*indica*, P1), while Milyang352 (*japonica*, P2) exhibited a high ROL rate. Under the same conditions, nearly 93.2% of the mapping population, comprising 117 DH lines, exhibited an enhanced ROL level. It could then be speculated that an increase or decrease in ROL rate could cause significant changes in the CH_4_ emission level, considering that this event is accelerated with the flow of O_2_-mediated aerobic conditions in the rhizosphere.

### 3.2. Stress-Responsive Genes Present within qROL-2-1 QTL Denotes Iron Sulfide-Induced Changes in ROL, May Involve Active Cellular Signaling Networks and Transcriptional Regulation in Rice

Depicting the genetic factors associated with the control of stress in plants is crucial to unlocking our understanding of the mechanism underlying the adaptive response mechanism towards stress tolerance. Communication in plants helps understand the mechanism underlying the stress response at different levels, including the physiological processes that take place, biochemical reactions which are activated, and molecular components involved in the adaptive response towards stress tolerance. Within the *qROL-2-1* region, we could see genes encoding transcription factors (TF) such as TEOSINTE-BRANCHED1CYCLOIDEA/PCF7 (TCP7) TF. In plants, TCP genes belong to the bHLH (basic helix–loop–helix) dimerizing transcription factor family, known as plant-specific TFs that play roles regulating multiple aspects of plant growth and development, including the meristem growth and development [65,66,67]. In addition, Dai, et al. [68] suggested that a member of the MYB (myeloblastosis) TF family, *OsMYB2P-1*, is involved in the control of root architecture under phosphate treatment. Similarly, Gu et al. [69] proposed *OsMYB1* TF as being involved in the regulation of phosphate homeostasis and root development in rice. Furthermore, another transcription factor encoding gene, a member of the LIM (LIN-11, Isl1, and MEC-3) family domain, found in the same QTL region, was earlier reported to be involved in seed and tiller development [70].

Moreover, a gene encoding a defensing-like DEFL protein (*OsDEF7*), here associated with the defense response mechanism and located in the *qROL-2-1*, would play a role in the adaptive response towards FeS tolerance. Defensins, commonly known as antibacterial peptides, function in the innate immune system. Other reports have supported that *OsDEF7* or *OsAPF1* are involved in the defense mechanism against pathogenic fungi [71,72], cadmium efflux and allocation [73], and lipid-binding and phosphate interaction [74] in rice.

### 3.3. Other Candidate Genes with the Potential to Control Radial Oxygen Loss in Rice

Plant hormones such as cytokinin and auxin play a preponderant role in biological systems. Auxin signaling is key to plant development and root growth. Here, we have detected an *ARF8* TF (Os02g41800), an auxin responsive factor previously reported to play a key role in the auxin signaling pathway [49].

In higher plants, multigene families encode expansins [75,76]. Expansins are identified as cell wall loosening proteins that mediate acid-induced growth by catalyzing the loosening of plant cell walls without lysis of wall polymers [77]. A recent report by Che et al. [78] proposed that an Al-inducible expansin gene, *OsEXPA10* (located on chromosome 4, Os04g49410), which is regulated by a C2H2-type zinc finger transcription factor (ART1), could be involved in the root cell elongation of rice. Later on, Tan et al. [79] proposed *OsEXPA10* to coordinate the balance between rice development and biotic stress resistance. Meanwhile, overexpression of *OsEXPA7* (chromosome 3, Os03g60720) was shown to confer salt tolerance in rice [80]. In a converse approach, ZhiMing et al. [81] indicated that root hair-specific expansins, including *OsEXPA17* (chromosome 6, Os06g01920) and *OsEXPA30* (chromosome 10, Os10g39110) modulate root hair elongation in rice. Knowing that root hair growth requires intensive cell wall modification, we could then speculate that the identified expansin precursor gene (chromosome 2, Os02g42650) harbored by the QTL *qROL-2-1* would play an important role in the control of radial oxygen loss (ROL) in rice.

LIM domain, a cysteine- and histidine-rich motif, has been proposed to govern protein–protein interactions. Many proteins containing LIM domains previously identified have been associated with gene regulation and cell fate determination, and tumor formation, among other processes [82].

Thus, with regard to their annotated molecular functions and biological processes in which they are proposed to be involved, coupled with their previous characterizations or similarity with reported genes, the set of genes found in the *qROL-2-1* QTL, such as those encoding EXPA (expansin), ARF8 TF (auxin responsive factor), PLIM2a TF, MYB21 TF, TCP7 TF, NIP2 (aquaporin), cb5 (cytochrome b5-like heme domain-containing protein), and WBC8 (ABC transporter), among others, may play a forefront role in the regulation of ROL in the rhizosphere of rice roots, which in turn may help to maintain at a low level the activity of methanogens and lower CH_4_ generation during rice cultivation. In addition, the presence of stress-responsive genes, including *OsDEF7* (defensin), *OsPSKR1* (phytosulfokine receptor precursor), *OsLRR2* (cysteine-rich domain containing protein), *OsTRX* (thioredoxin), and *OsRRA8* (response regulator 11) in the *qROL-2-1* would imply that plants exposed to FeS experience stressful conditions, and their interplay with ROL-related genes may contribute to a balanced root growth.

Therefore, with regard to the recorded transcript accumulation patterns of *qROL-2-1*-related genes associated with ROL in rice (Figure 5A–J), upregulation of the majority of the target genes in Milyang352, a high ROL cultivar (Figure 5A–C,F,J,K), and an increased transcript accumulation level of *OsDEF7*, *OsNIP2*/*OsLsi1*, and *Oscb5* (Figure 5E,H,J) in 93-11, a low ROL cultivar, coupled with *OsEXPA* and *OsNIP2* being significantly downregulated by FeS over time, we could speculate that the first group of genes could play a role in the promotion of ROL in rice, while the second group could be suggested as important genes for the control of oxygen flux level in the roots of rice during FeS-induced stress.

## 4. Materials and Methods

### 4.1. Plant Materials, Growth Conditions, and FeS Medium Preparation

The mapping population comprised 117 doubled haploid (DH) rice lines developed through another culture and their parental lines, 93-11 (P1) and Milyang352 (*indica* and *japonica* subspecies, respectively). Prior to sowing, rice seeds were surface sterilized using nitric acid (HNO_3_) 0.7% for 15 min, followed by rinsing three times with distilled water, and the excess water was dried on a sterile paper towel. Then, two seeds per line per plate were sown in triplicate on agar medium modified with iron (II) sulfide (FeS).

The FeS medium was prepared as previously described by Fleck et al. [83], with slight modifications. Briefly, 0.8% (8 g L^−1^) Bacto^TM^ Agar (Lot 0283880, Becton, Dickinson and Company, Sparks, MD 21152, USA) was mixed with distilled water, and autoclaved at 121 °C for 15 min to solubilize the agar, and cooled down to about 50 °C at room temperature. Then 1.4 g L^−1^ iron (II) sulfate heptahydrate (FeSO_4_ 7H_2_O) (CAS: 7782-63-0, Sigma-Aldrich, St. Louis MO 63103, USA) and 0.32 g L^−1^ sodium sulfide (Na_2_S) (CAS: 1313-84-4, Sigma-Aldrich, St. Louis MO 63103, USA) were added simultaneously, and homogenized by hand shaking whereupon a black FeS precipitation was formed [84]. The color of the medium turned black, which indicates that the reaction of FeSO_4_ with Na_2_S took place. Finally, the solution was buffered by addition of 0.5 g L^−1^ calcium carbonate (CaCO_3_) and adjusted to pH 6.0. Then, 50 mL was distributed to sterile square culture plates (12 cm × 12 cm × 2 cm for length, width, and height, respectively), and the plates were kept until the color of the medium changed from black to yellowish. To visualize the oxidation power, surface sterilized rice seeds with hulls were sown on FeS medium, and the plates were placed in a growth chamber with 16 h of light and 8 h of dark periods at ±25 °C.

To allow proper root elongation and development in the direction of the length of the plate (gravitropism) on the agar medium, plates were placed with an inclination of about 60 degrees and exposed to the light source. The root length, oxidation area, and ROL were measured two weeks after sowing. 

### 4.2. Determination of Root Oxidized Area, Radial Oxygen Loss, and Root Length

To determine the root area and the radial oxygen loss percentage, square cell culture plates containing two seedlings with well-developed roots were scanned using Fuji Xerox Apeosport-V C3373CG, Tokyo, Japan. Then, scanned images were analyzed using the image processing software ImageJ version 1.46 (Wayne Rasband, National Institute of Mental Health, Bethesda, Maryland, USA) [85]. Briefly, a target image was opened from its location on the computer (in the title bar, click on File → Open (Ctrl + O)). Prior to analysis, the image samples were changed to 32-bit image format, which denotes the overall pixel intensity (in the title bar, click on image → type → 32-bit), followed by the adjustment of the image threshold (in the title bar, click on image → adjust → threshold), which is applied to all analyzed image samples. Then, the oxidized area was selected and the analysis tab was used to quantify. The RA and ROL were expressed in percentage.

The root length was measured after recording the ROL and RA data. Briefly, seedlings were carefully removed from the agar plates, followed by the removal (by absorbing) of the medium surrounding the roots with a paper towel. Then, the length of the roots was measured from the collar of the seedling to the tip of the root, after carefully stretching out alongside a 30-cm ruler on a bench.

### 4.3. Frequency Distribution, Box Plots, and Principal Component Analysis

The frequency distribution and box plots of the evaluated phenotypic traits in response to FeS were generated using GraphPad Prism software (version 7.00, © 2022–2016 GraphPad, San Diego, CA, USA). The normality of the distribution was assessed using the Shapiro–Wilk W-statistic for the test of normality in IciMapping (version 4.1.0.0, 2016, Chinese Academy of Agriculture Sciences, Beijing, China). In addition, the pairwise kinship matrix was generated using the GAPIT function: *my_GAPIT <−GAPIT (Y = myY, G = myG, Model.selection = TRUE, SNP.MAF = 0.05)* in RStudio (version 1.2.5042, © 2022–2020 RStudio, Inc., Boston, MA, USA). The cluster analysis was performed using the R packages *ggplot2*, *tidyverse*, *cluster*, and *fviz_cluster (fviz_nbclust (y, kmeans, method = “silhouette”; final <−kmeans (y, 2, nstart = 25); fviz_cluster (final, data = y, labelsize = 8)* R function. The principal component analysis (PCA) was performed using the R package *ggplot2* and functions *myPr*, *plot*, and *biplot* [86,87].

### 4.4. Construction of Linkage Mapping and QTL Analysis

We used 240 markers, including 105 KASP markers [88] and 135 Fluidigm markers [14] specific for detecting polymorphism between *japonica* subspecies, and the phenotypic data of the mapping population consisting of 117 double haploid (DH) lines and their parental lines (93-11, P1 and Milyang352, P2) were used to perform QTL analysis, to identify putative QTLs controlling radial oxygen loss (ROL) in the aerenchyma area of rice roots. Prior to performing the QTL analysis, the experiments were repeated three times to validate the phenotype.

The QTL analysis was performed and the linkage maps constructed with IciMapping software v. 4.1.0.0, for a biparental population using position mapping and Kosambi mapping functions [89]. The permutation (1000 times) parameters, which explains the probability for detecting statistically significant QTLs associated with the target traits, were selected.

The rice genome annotation database, publicly available online browser (http://rice.plantbiology.msu.edu/, accessed on 22 November 2021), was used to identify candidate genes pooled from the detected QTL *qROL-2-1* (745,184bp, Chr2:24954729..25699913) region flanked by KASP markers KJ02_047 (left, closest marker) and ad02011485 (right marker) on chromosome 2, associated with the control of ROL in rice.

### 4.5. qPCR Validation of Putative ROL Candidate Genes in Response to FeS

Prior to performing qPCR, surface sterilized and pre-germinated seeds were sown in a 50-well tray containing an enriched soil (Doobaena Plus, Nong Kyung Ltd., Yeongcheon-si, Korea) for 14 days. Then, 14-day-old seedlings were removed carefully after loosening the soil around the roots, and the roots were gently washed with tap water. The excess water was absorbed with a paper towel and the roots were immediately dipped into the FeS solution (10 mL in each 50 mL falcon tube) in triplicate. Distilled water was used as the control treatment.

Total RNA was extracted from root samples using the RNeasy^®^ Plant Mini Kit (Cat. 74904, Lot 160037043, QIAGEN GmbH, QIAGEN Strasse1, 40724 Hilden, Germany) according to the manufacturer’s instructions. In essence, frozen root samples with liquid nitrogen were crushed to fine powder with a pestle in a ceramic mortar and put in 2 mL Eppendorf tube (e-tube), and 450 μL of lysis buffer RLT (containing 10 μL β-mercaptoethanol (β-ME) per 1 mL of buffer RLT) was added and mixed by vigorously vortexing for few seconds. The lysate was transferred to a QIAshredder spin column (liliac) placed in a 2 mL collection tube, followed by centrifugation at 12,000 rpm for 2 min (top bench centrifuge). The supernatant of the flowthrough was transferred to a new 1.5 mL e-tube (Step 1). Then, to the mixture from Step 1, 0.5 volume of ethanol (96–100%) was added, followed by mixing by pipetting up and down. A total of 650 μL was transferred to an RNeasy Mini spin column (pink) with a 2 mL collection tube, followed by centrifugation for 15 sec at 12,000 rpm, and 700 μL of buffer RW1 was added after discarding the flowthrough. The tubes were then centrifuged for 15 sec at 12,000 rpm. Immediately after discarding the flowthrough, 500 μL of buffer RPE was added to the RNeasy spin column, and centrifuged for 2 min at 12,000 rpm. Empty spin columns with collection tubes were centrifuged for 1 min at 12,000 rpm to dry the membrane, and the spin columns were placed on new 1.5 e-tubes. Finally, 50 μL of nuclease-free water was added to the spin column, and the tubes were centrifuged for 1 min at 12,000 rpm to elute the RNA (this step was repeated twice using the flowthrough). RNA samples were stored at −20 °C.

For cDNA synthesis, 1 μg of RNA was used as a template, and the ProtoScript^®^ II First Strand cDNA Synthesis Kit (New England BioLabs Inc., NEB Labs, MA, USA) was employed according to the manufacturer’s instructions (Step 1: pre-heat the mixture containing the calculated volume in μL for 1 μg RNA sample and 1 μL Oligo dT at 65 °C for 5 min. Step 2: add 2 μL of 10 × ProtoScript II Enzyme Mix to the mixture from Step 1, and adjust the reaction volume to 20 μL with nuclease-free water. Run PCR at 42 °C for 1 h). The newly synthesized cDNA (stored at −20 °C) was then used as a template for qPCR (quantitative real-time polymerase chain reaction) to investigate the changes in the transcript accumulation of selected putative candidate genes.

The qPCR reaction mixture comprised 10 μL of amfiSure qGreen qPCR Master Mix (2×) containing low ROX (Q5603, GenDEPOT CORPORATION, Barker, TX 77413, USA), 1 μL of cDNA, and 1 μL of each forward and reverse primers in a total reaction volume of 20 μL, including a no-template control (NTC). A three-step reaction including polymerase activation at 95 °C for 5 min, denaturation at 95 °C for 20 s, annealing at 60 °C for 30 s, and extension at 72 °C for 30 s was performed in a real-time PCR machine (QuantStudio^TM^ Design and Analysis Software v. 1.3, Applied Biosystems, Thermo Fisher Scientific, Seoul, Korea), with a total of 40 reaction cycles, and the data were normalized to the relative expression of rice Actin1. The list of primers used in the study is attached as Appendix A.

## 5. Conclusions

Depicting the molecular basis for radial oxygen loss (ROL) in plants is key to developing crop varieties with an improved ability to lower microbial activity in the rhizosphere of rice roots, such as methanogens, which may contribute to reducing greenhouse gas (GHGs) emissions from agriculture. This study identified a novel QTL (*qROL-2-1*) associated with the control of ROL in rice. The genes located in this QTL have interesting annotated molecular functions, and are proposed to be involved in various biological processes and plant stress response mechanisms. Among them, the transcriptional regulation of 11 genes was investigated by qPCR in response to iron (II) sulfide (FeS) in the roots of 93-11 (P1 with low ROL rate) and Mlyang352 (P2 with high ROL rate). Of this number, genes encoding TEOSINTE-BRACHED1CYCLODOIDEA/PCF7 (*OsTCP7*), thioredoxin (*OsTRX*), myeloblastosis TF (*OsMYB21*), auxin responsive factor (*OsARF8*), *OsWBC8* (ABC transporter), and *OsLRR2* (leucine-rich repeat) (Set 1) exhibited a sustained increase in their transcript accumulation in Milyang352 compared to 93-11. In contrast, genes encoding defensin (*OsDEF7*), *Oscb5* (cytochrome b5-like heme domain-containing protein), and LIM domain (*OsPLIM2a*) (Set 2) exhibited an increase in their transcript accumulation in 93-11 compared to Milyang352. Interestingly, *OsEXPA* (encoding expansions) and *OsNIP2* (encoding aquaporin) (Set 3) were differentially regulated between 93-11 and Milyang352. Therefore, all results put together suggest that the genes in set 1 would act as positive regulators of ROL, while those in sets 2 and 3 are suggested to be involved in the control of oxygen flux level in rice roots under FeS treatment conditions, which in turn may help to maintain at a low level the activity of methanogens and lower methane (CH_4_) generation during rice cultivation.

Moreover, the presence of stress-responsive genes, such as *OsPSKR1* (phytosulfokine receptor precursor) and *OsRRA8* (response regulator 11) in the *qROL-2-1* would imply that plants exposed to FeS experience stressful conditions, and their interplay with other ROL-related genes may contribute to ensuring a balanced root growth.

## Figures and Tables

**Figure 1 plants-11-00788-f001:**
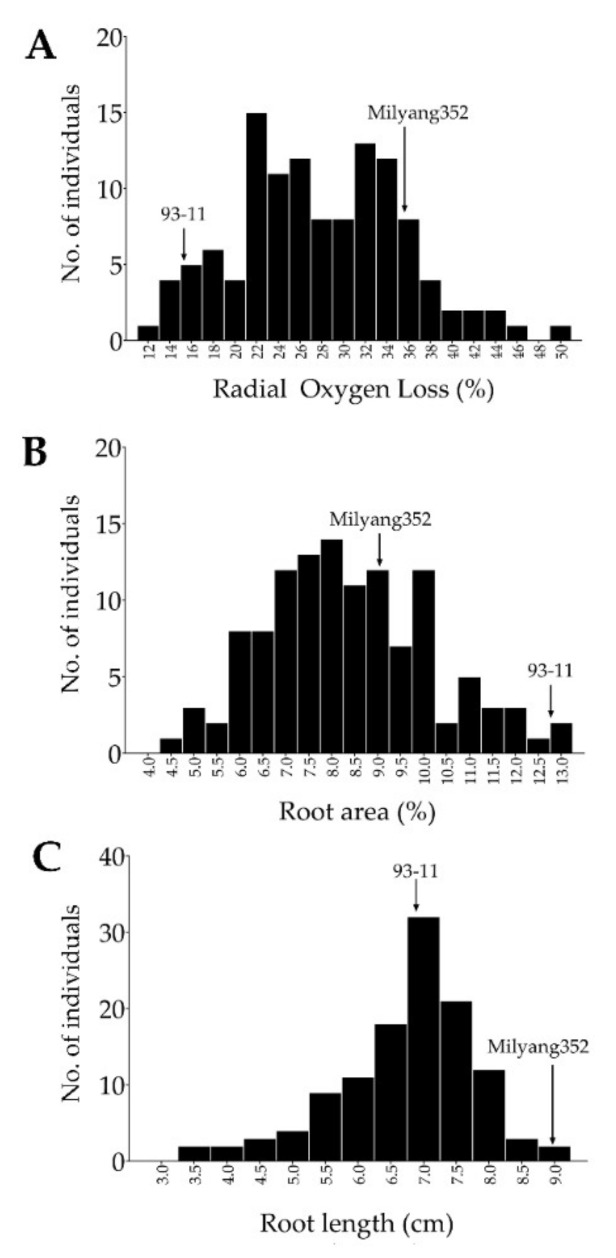
Frequency distribution of traits. (**A**) Frequency distribution of radial oxygen loss (ROL) of a doubled haploid (DH) population grown on FeS. (**B**) Frequency distribution of root area (RA), and (**C**) frequency distribution of root length (RL) under the same conditions.

**Figure 2 plants-11-00788-f002:**
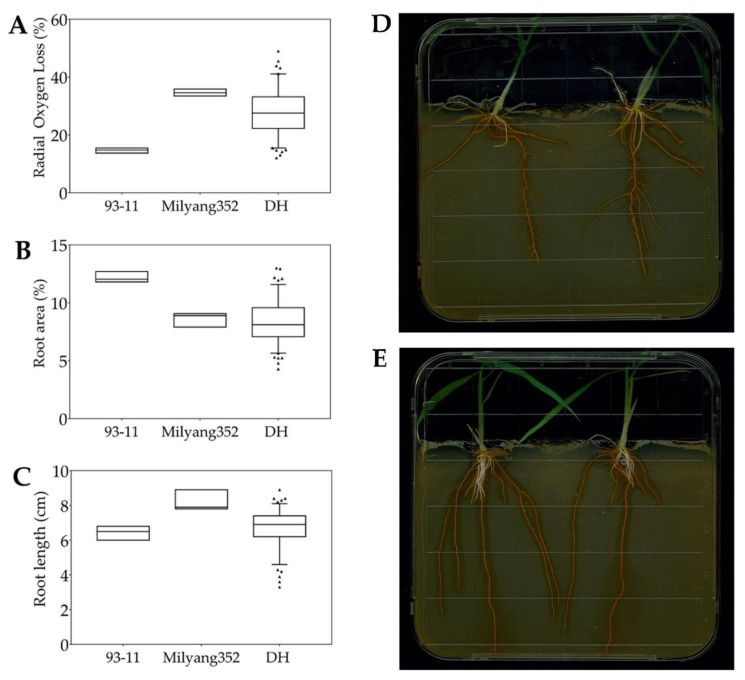
Genotype–phenotype correlation of the mapping population and their parental lines under FeS treatment. (**A**) The box plot shows the effects of FeS application on ROL rate of 93-11 (P1, *indica*), Milyang352 (P2, *japonica*), and that of a doubled haploid (DH) rice population (*n* = 117), (**B**) box plot showing the root area (RA), and (**C**) root length (RL) of the same mapping population, under the same conditions, (**D**) ROL phenotypes of 93-11 and (**E**) Milyang352. Pictures were scanned 14 days after sowing on FeS media.

**Figure 3 plants-11-00788-f003:**
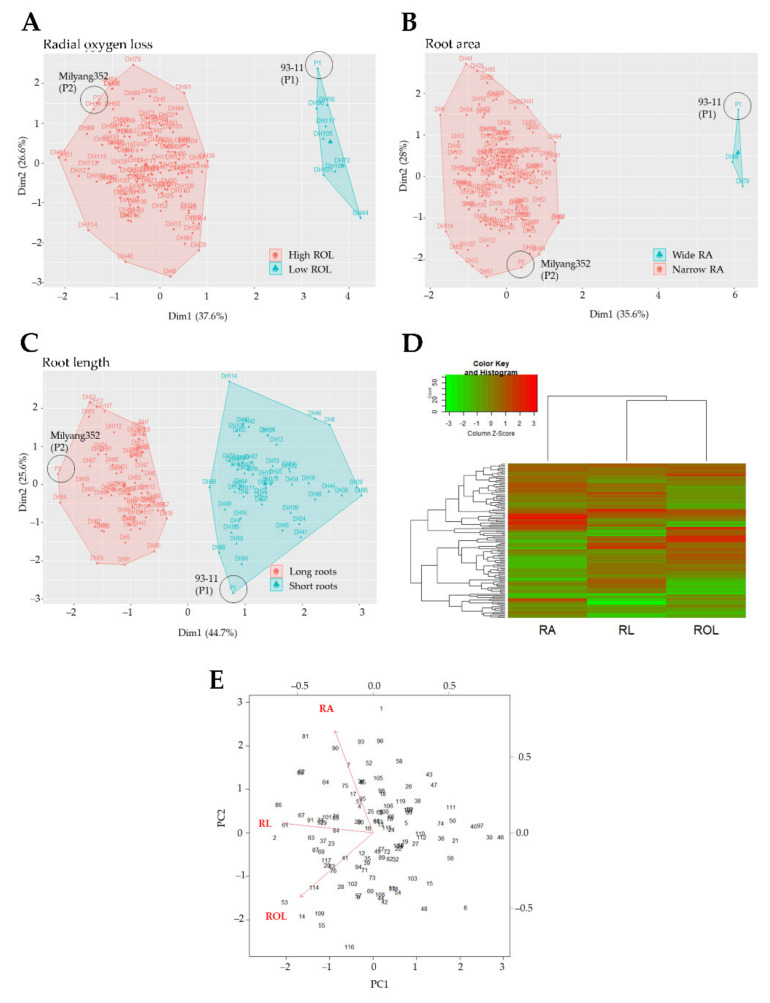
Principal component analysis results. Cluster plots displaying the distinctive phenotypes of parental lines (93-11, P1 and Milyang352, P2) and the grouping of the mapping population based on radial oxygen loss (ROL) rate (**A**), (**B**) root area (RA), and (**C**) root length (RL). (**D**) Heat map with dendrograms indicating the affinity between traits in response to FeS treatment, (**E**) PCA indicating the correlation between the analyzed traits on the mapping population.

**Figure 4 plants-11-00788-f004:**
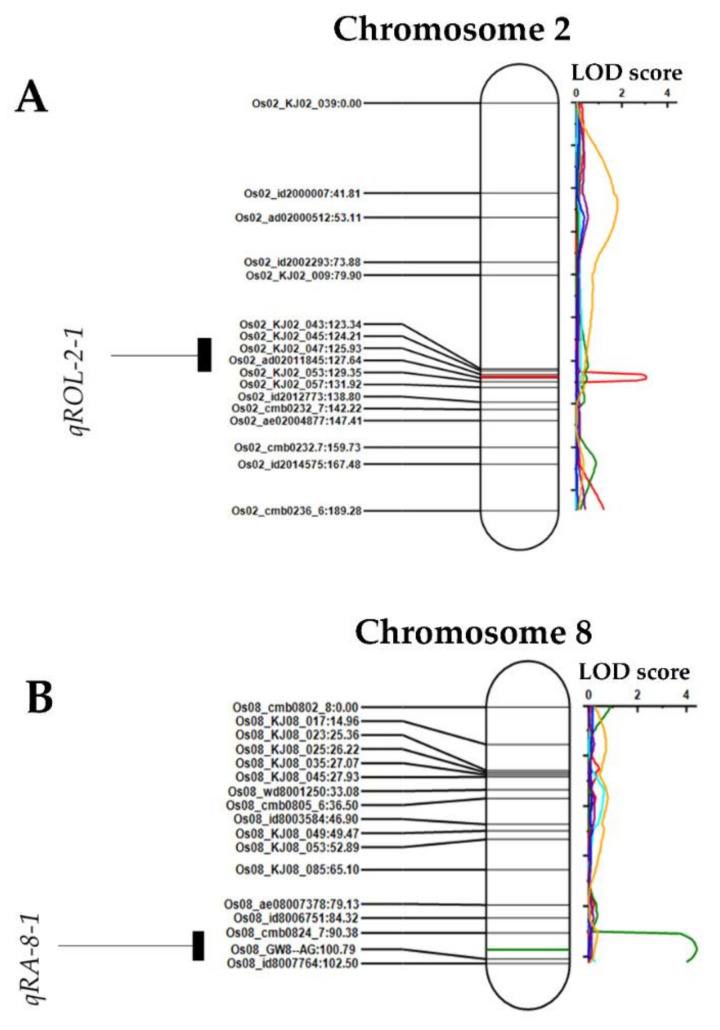
Linkage maps and QTLs associated with radial oxygen loss (ROL) in rice seedlings. (**A**) A unique QTL (*qROL-2-1*, 127 cM indicated by the red line inside the chromosome and the red LOD peak) associated with ROL in rice was detected on chromosome 2. (**B**) One QTL (*qRA-8-1*, 97 cM indicated by the green line inside the chromosome and the green LOD peak) associated with root area under FeS stress was mapped on chromosome 8.

**Figure 5 plants-11-00788-f005:**
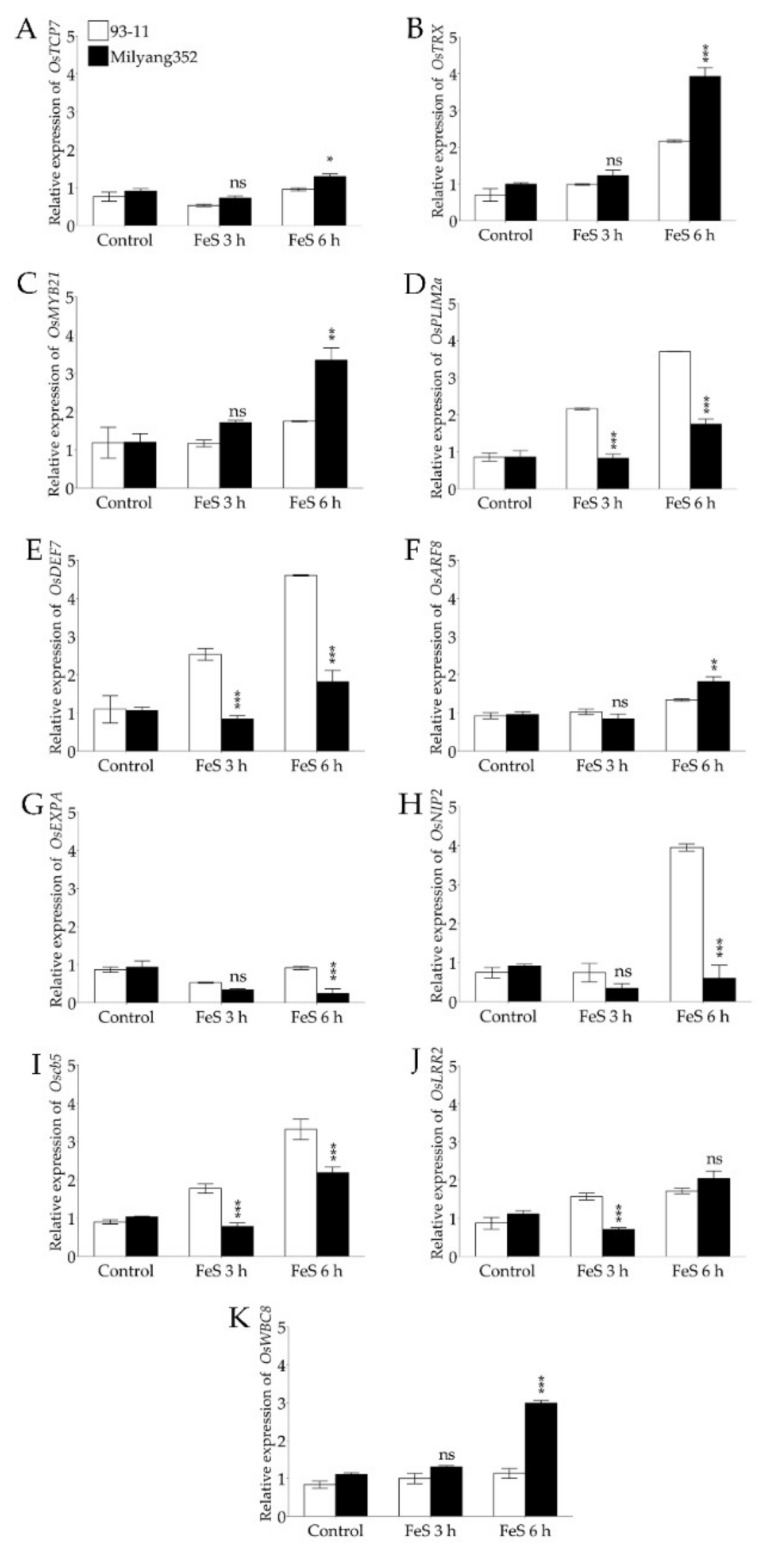
Over time transcript accumulation patterns of *qROL-2* related genes associated with radial oxygen loss (ROL) in rice in response to FeS treatment. Relative expression of (**A**) *OsTCP7*, (**B**) *OsTRX*, (**C**) *OsMYB21*, (**D**) *OsPLIM2a*, (**E**) *OsDEF7*, (**F**) *OsARF8*, (**G**) *OsEXPA*, (**H**) *OsNIP2*, (**I**) *Oscb5*, (**J**) *OsLRR2*, and (**K**) *OsWBC8* in the roots of 14-day-old rice seedlings of 93-11 (P1, *indica* low ROL cultivar) and Milyang352 (P2, *japonica* high ROL cultivar) exposed to iron sulfide (FeS) over time. Data are mean values of triplicate. Error bars are mean values ± SE. **** p* < 0.001, *** p* < 0.01, ** p* < 0.05, *ns* non-significant.

**Figure 6 plants-11-00788-f006:**
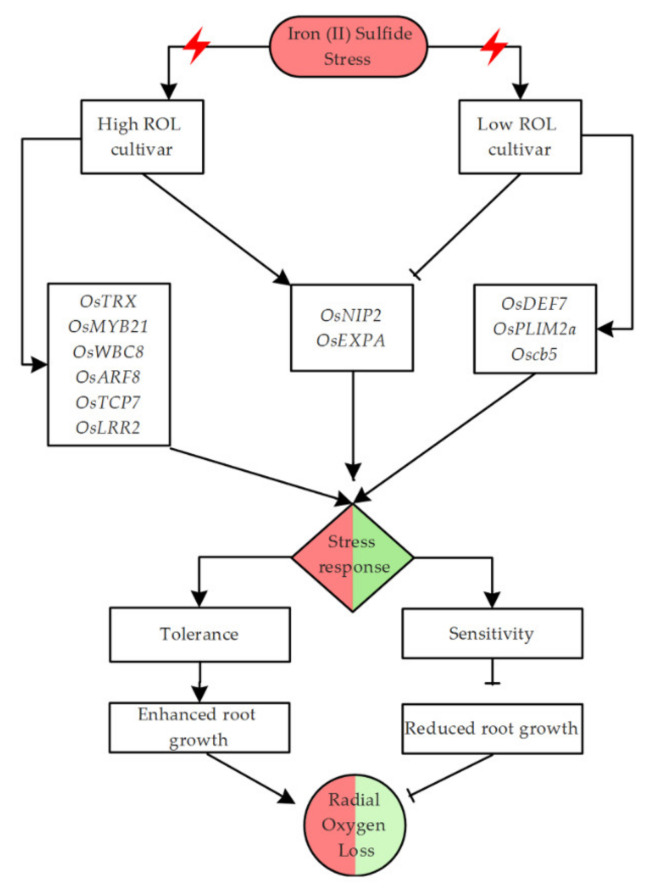
Hypothetical signaling model in response to FeS-mediated radial oxygen loss (ROL) in rice. When plants are exposed to an environmental stimuli, they activate signaling cascades as part of the defense mechanism, which include the induction of a wide range of stress-responsive genes, including transcription factors. The interplay between the genes and other components of the adaptive response mechanism determines the degree of the tolerance. Iron (II) sulfide (FeS) is shown here to significantly increase the transcript abundance of *OsTRX*, *OsMYB21*, *OsWBC8*, *OsARF8*, *OsTCP7*, and *OsLRR2* in Milyang352 (high ROL cultivar). In the same way, *OsDEF7*, *OsPLIM2a*, and *Oscb5* is highly induced in 93-11 (low ROL cultivar), while *OsNIP2* and *OsEXPA* were differentially regulated. In addition, exogenous application of FeS is shown to cause reduction in root growth of 93-11 but not in Milyang352, which resulted in different stress response levels and ROL patterns. Continuous lines with an arrow at their tips indicate a positive or induction (of gene expression or roots growth), whereas, continuous lines with a perpendicular bar at one end suggest a negative regulation or inhibition (of gene expression or roots growth). Red/green color schemes in the shapes indicates high/low response or ROL level, respectively. This model was designed using ConceptDraw PRO v. 10.3.2.114 (© 2022–2022 CS Odessa Corp., San Jose, CA 95113-1116, USA) and edited in Adobe^®^ Photoshop^®^ CS6 (v. 13.0.1 ×32, © 2022–2012 Adobe Systems Incorporated, USA).

**Table 1 plants-11-00788-t001:** Detected QTLs associated with radial oxygen loss and root area of rice.

Trait (a)	QTL (b)	Chr (c)	Position (cM) (d)	Left Marker (e)	Right Marker (f)	LOD (g)	PVE (%) (h)	Add (i)	LeftCI (j)	RightCI (k)
ROL	*qROL-2-1*	2	127	Os02_KJ02_047	Os02_ad02011845	3.038	11.6082	2.5317	125.5	129.5
RA	*qRA-8-1*	8	97	Os08_cmb0824_7	Os08_GW8--AG	4.394	15.9468	0.7613	90.5	102

(a) Rice traits for which QTLs were identified: radial oxygen loss (ROL) and root area (RA). (b) Detected QTL names; (c) chromosome number; (d) absolute position of the QTL from top of the linkage map in centimorgan (cM); (e) left flanking markers; (f) right flanking markers; (g) logarithm of the odds (LOD) scores; (h) phenotypic variation explained (PVE) by the QTLs, expressed in percentage; (i) additive effect: the positive value shows that the allele from 93-11 (P1) increased the trait value; (j,k) left and right marker positions in cM.

**Table 2 plants-11-00788-t002:** Candidate genes harbored by the *qROL-2-1* associated with radial oxygen loss in rice.

No	Gene Name	MSU ID	Annotation	Biological Process	Molecular Function	Cellular Component
1	*ARF8* TF	Os02g41800	Auxin response factor, putative, expressed; ARF family protein	Multicellular organismal development; response to endogenous stimulus	RNA binding; sequence-specific DNA binding transcription factor activity	Nucleus
2	*OsLsi1*/ *NIP2*	Os02g41860	Aquaporin protein, putative	Transport; cell differentiation; response to endogenous stimulus	Transporter activity	Membrane
3	*OsPSKR1*	Os02g41890	Phytosulfokine receptor precursor	Response to stress; signal transduction; protein modification process	Kinase activity; receptor activity; binding; signal transducer activity	Cell
4	*OsDEF7/CAL1/* *OsAFP1*	Os02g41904	Defensin-like DEFL family	Response to stress; defense response	Enzyme regulator activity	Cell wall; plasma membrane
5	*OsWBC8*	Os02g41920	White-brown complex homolog protein 8, ABC-2 type transporter domain containing protein	Transport; catabolic process; nucleic acid metabolic process	Hydrolase activity; transporter activity	Membrane
6	*OsRR11*/*OsRRA8*	Os02g42060	Response regulator receiver domain containing protein; A-TYPE response regulator 11	Signal transduction; response to abiotic stimulus; response to auxin and cytokinin stimuli; two-component signal transduction system (phosphorelay)	Protein binding; signal transducer activity	Nucleus
7	*OsWee1*	Os02g42110	Wall-associated receptor kinase-like 22 precursor; cell cycle-associated protein kinase	Protein modification process; cell surface receptor linked signal transduction; calcium ion binding;	Kinase activity; protein binding; negative regulation of cell division; protein serine/threonine kinase activity	Plasma membrane
8	*OsARF*	Os02g42134	ARF GTPase-activating domain-containing protein	Signal transduction; regulation of ARF GTPase activity	ARF GTPase activator activity; phospholipid binding; zinc ion binding; metal ion binding	Cytoplasm
9	*OsWAK14*	Os02g42150	Wall-associated kinase, receptor-like protein kinase; EGF-like calcium-binding domain containing protein.	Protein modification process	Kinase activity; calcium ion binding; polysaccharide binding	Plasma membrane
10	*Lipase*	Os02g42170	Phospholipase, putative, expressed; lipase, class 3 domain containing protein	Response to stress; response to biotic stimulus; cellular process; lipid metabolic process	Hydrolase activity; triglyceride lipase activity	Cytoplasm; plastid
11	*OsDLN61*	Os02g42200	DLN REPRESSOR 61, B-block binding subunit of TFIIIC domain containing protein	Transcription initiation from RNA polymerase III promoter	5S class rRNA transcription	
12	*OsUGT*	Os02g42280	UDP-glucoronosyl/UDP-glucosyl transferase family protein	metabolic process	Transferase activity	Cell wall
13	*OsE2*/ RWD	Os02g42314	Ubiquitin-conjugating enzyme/RWD-like domain containing protein	Fatty acid beta-oxidation; protein modification process; transport	Protein binding; ligase activity; acid-amino acid ligase activity	
14	*OsTCP7*	Os02g42380	TCP domain containing protein	Biosynthetic process	Protein binding; sequence-specific DNA binding transcription factor activity	Nucleus
15	*OsLRR2*	Os02g42412	Leucine-rich repeat 2, cysteine-containing subtype containing protein	RNA-dependent DNA biosynthetic process	RNA-directed DNA polymerase activity; RNA binding	
16		Os02g42520	Dehydrogenase	Metabolic process	Binding; catalytic activity	plastid
17	*OsFTR*	Os02g42570	Ferredoxin-thioredoxin reductase, variable chain	Generation of precursor metabolites and energy; photosynthesis	Catalytic activity	Plastid
18	*OsBRO1*	Os02g42580	Vacuolar protein-sorting protein bro1	-	-	Cytosol
19	*OsAP2* TF	Os02g42585	AP2 domain containing protein	Biosynthetic process; nucleic acid metabolic process	DNA binding; sequence-specific DNA binding transcription factor activity	Nucleus; cytoplasm
20	*OsWD*-40	Os02g42590	WD-40 repeat family protein, putative, expressed	Signal transduction	Signal transducer activity	Intracellular; plasma membrane
21		Os02g42600	Double-stranded RNA binding motif containing protein	Response to stress; signal transduction; response to abiotic stimulus	Hydrolase activity; RNA binding; protein binding;	Nucleus; intracellular
22	*OsEXPA*	Os02g42650	Expansin precursor, putative, expressed	Anatomical structure morphogenesis; cell growth; cellular process	-	-
23	*OsMLD*/ *DES1*	Os02g42660	Degenerative spermatocyte homolog 1, lipid desaturase/migration-inducing gene 15 protein/sphingolipid delta 4 desaturase protein	Sphingolipid biosynthetic process; plant-type cell wall modification; fatty acid biosynthetic process; oxidation-reduction process	Sphingolipid delta-4 desaturase activity; oxidoreductase activity, acting on paired donors, with incorporation or reduction of molecular oxygen	Integral component of membrane
24	Os*DUF1771*	Os02g42670	DUF1771 domain containing protein	Regulation of gene expression, epigenetic	-	-
25		Os02g42690	Zinc finger, C3HC4 type domain containing protein	Transport; protein modification process; protein metabolic process	Catalytic activity; binding	Membrane; endoplasmic reticulum;
26	*OsTRX*	Os02g42700	Thioredoxin	Response to stress; metabolic process	Enzyme regulator activity	Membrane; plastid; thylakoid
27	*Oscb5*	Os02g42740	Cytochrome b5-like Heme/Steroid binding domain containing protein	Binding		Membrane; plastid; thylakoid; nucleus
28	*Lectin*	Os02g42780	Lectin receptor-type protein kinase	Metabolic process; cellular process	Kinase activity	Plasma membrane
29		Os02g42790	Short-chain dehydrogenase/reductase	Metabolic process; response to abiotic stimulus;	Catalytic activity; response to stress; oxidoreductase activity	Cytosol; plasma membrane
30		Os02g42810	Oxidoreductase, short-chain dehydrogenase/reductase family domain containing protein	Metabolic process	Catalytic activity; binding	Cell wall
31	*OsPLIM2a*	Os02g42820	LIM domain protein, putative actin-binding protein, and transcription factor	Cellular component organization; cellular process	Protein binding	Cytoskeleton
32	*OsMYB21*	Os02g42850	MYB family transcription factor	Biosynthetic process; nucleobase, nucleoside, nucleotide, and nucleic acid metabolic process	Sequence-specific DNA binding transcription factor activity	

## Data Availability

Not Applicable.

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
