# Peer review of "Novel QTL Associated with Aerenchyma-Mediated Radial Oxygen Loss (ROL) in Rice (Oryza sativa L.) under Iron (II) Sulfide"

_plants, 2022, doi:10.3390/plants11060788_

Round 1

Reviewer 1 Report

 1- No details are provided on algorithm or software used for QTL identification  

2- No  validation of the annotated genes to confirm their contribution/ Involvement in ROL. Their is a need to confirm that qROL-2 harbors genes proposed to be involved in stress signaling, defense response mechanisms, and transcriptional regulation. some validation of these potential genes will narrow down and identify real candidate genes involved in ROL.

3- Many typing errors and unnecessary brackets in manuscript, thus critical editing is needed

Author Response

Title: Novel QTL Associated with Aerenchyma-Mediated Radial Oxygen Loss (ROL) in Rice (Oryza sativa L.) under Iron (II) Sulfide

Manuscript ID plants-1539996

Journal Plants

Section Plant Molecular Biology

Special Issue Development of System to Conventional and QTL Breeding for Rice Cultivars

Point by point reply to reviewers’ comments

We are thankful to the editorial team and anonymous reviewers for their time dedicated to this manuscript. We sincerely appreciate the comments, which helped us improved significantly the content and the quality of the manuscript. We would like to specify that all changes in the manuscript in the first round of revision were highlighted green. We hope that the manuscript in the present form is suitable for publication in the journal.

Reviewer 1

The authors would like to thank the reviewer for his valuable comments and suggestions which have helped us improve significantly the manuscript.

1- No details are provided on algorithm or software used for QTL identification  

We apologize for the inconvenience. We have added the necessary description of QTL analysis in the materials and methods section (lines 505–515).

2- No validation of the annotated genes to confirm their contribution/ Involvement in ROL. There is a need to confirm that qROL-2 harbors genes proposed to be involved in stress signaling, defense response mechanisms, and transcriptional regulation. Some validation of these potential genes will narrow down and identify real candidate genes involved in ROL.

We appreciate the suggestion made by the reviewer. We have included new data, described, and discussed the results of the validation by qPCR of selected genes harbored by the qROL-2-1 QTL in the manuscript (281–345; 433–441; 516–561).

3- Many typing errors and unnecessary brackets in manuscript, thus critical editing is needed

We apologize for the inconvenience. We have tried to improve the manuscript for the typos and other errors.

Reviewer 2 Report

The manuscript reports rice QTL for radial oxygen loss, an interesting root trait potentially associated with greenhouse gas emission in rice paddy. I think the major limitation of this study is that the QTL analysis on such complex quantitative trait was conducted only once without replications. Repeating the experiments using two-week old seedlings with a small DH population (n=117) should not be too difficult or time-consuming, so I highly recommend the authors repeat the experiments few more times (preferably two more times) to search stable QTLs. Please also check the comments below to further improve the manuscript.

Line 47: I am not sure if this statement is true. Rice is used not only for human consumption but also for animal feeding.

Line 62: “environmental stresses” or “abiotic stresses” instead of “environmental cues”

There are many sentences missing proper references across the manuscript (e.g., line 67, 70, 76, 104, 374).

Line 82-84: The sentence begins with introducing two different types of aerenchyma, but actually describes only one type.

Line 107-108: I think this section requires more explanation with proper references regarding the use of specific agar medium with iron sulfide.

Line 121-129: The criteria used to divide the mapping population into two groups (P1-like and P2-like) for the three traits (ROL, root area and root length) are unclear. All three traits are quantitative with continuous variation and I don’t think it is proper to classify the mapping population simply into two groups using arbitrary criteria (there is a similar concern regarding line 262). In addition, Figure 1 and 2 seem to be redundant presenting the same data in two different formats and I think one of them can be omitted from the main manuscript. For Fig 1A, 1B, and 1C, axis information is labeled incorrectly – x axis, not y, should be labeled with each trait.

Line 141-151: What is the point of conducing PCA and clustering analysis with only three traits and describing trait correlations using the PCA result? I would suggest conducting simple correlation analysis instead and describing trait correlations directly using actual correlation coefficients.

Line 159: Please mention specifically how many of each marker (KASP and Fluidigm) were used for genotyping. What was the total linkage map length in cM?

Line 162-167: These sentences should be rephrased. The flanking markers and additive effects are described for only one QTL while dealing with two separate QTLs. Please also add more information regarding the QTL for root length along with the LOD threshold used for declaring significant QTLs.

Figure 4: The legend should provide information on different colors used in the LOD graph.

Section 3.2: Be consistent with the QTL nomenclature. qROL-2 or qROL-2-1?

Line 354-360: Most information described here is redundant with line 333-338. In Section 4.1, more information is required regarding experimental design. How many seeds were sown per each line and each culture plate? How many biological replications did you use per line? Is it correct that the height of the culture plate is only 2 cm while the root length measured was in the range of 3 – 9 cm?

Section 4.2: Information on root length measurement should be also included. Was it measured directly, or was it also image-based data?

Section 4.3: More information is required on linkage map construction and QTL analysis. What software was used? What were the functions used for linkage map and QTL? What is the LOD threshold used?

Author Response

Title: Novel QTL Associated with Aerenchyma-Mediated Radial Oxygen Loss (ROL) in Rice (Oryza sativa L.) under Iron (II) Sulfide

Manuscript ID plants-1539996

Journal Plants

Section Plant Molecular Biology

Special Issue Development of System to Conventional and QTL Breeding for Rice Cultivars

Point by point reply to reviewers’ comments

We are thankful to the editorial team and anonymous reviewers for their time dedicated to this manuscript. We sincerely appreciate the comments, which helped us improved significantly the content and the quality of the manuscript. We would like to specify that all changes in the manuscript in the first round of revision were highlighted green. We hope that the manuscript in the present form is suitable for publication in the journal.

Reviewer 2

The manuscript reports rice QTL for radial oxygen loss, an interesting root trait potentially associated with greenhouse gas emission in rice paddy.

I think the major limitation of this study is that the QTL analysis on such complex quantitative trait was conducted only once without replications. Repeating the experiments using two-week old seedlings with a small DH population (n=117) should not be too difficult or time-consuming, so I highly recommend the authors repeat the experiments few more times (preferably two more times) to search stable QTLs. Please also check the comments below to further improve the manuscript.

The authors appreciate the concerns raised by the reviewer, and agree on the premises. We would like to indicate that the experiments were repeated three times using the same population (DH lines) and under the same conditions to validate the phenotype prior to performing the QTL analysis. Therefore, the results presented in the study is the outcome of these experiments.

Line 47: I am not sure if this statement is true. Rice is used not only for human consumption but also for animal feeding.

We appreciate the concern raised by the reviewer. We agree with the suggestion that not 100% of rice production and by-products goes to human nutrition. Therefore, we have modified the statement as follows: Since its domestication, rice remains the only cereal crop mainly cultivated for human consumption, and half of the world’s population is said to depend on rice as a staple food, with a small portion of global rice production (grains) used for animal feeding. [1-3] (lines 53–54)

Line 62: “environmental stresses” or “abiotic stresses” instead of “environmental cues”

Line 67 (62 in previous version): we have replaced the wording “cues” with “stresses” as suggested.

There are many sentences missing proper references across the manuscript (e.g., line 67, 70, 76, 104, 374).

We apologize for the inconvenience. We added references accordingly throughout the manuscipt.

Line 82-84: The sentence begins with introducing two different types of aerenchyma, but actually describes only one type.

We described a bit more the lysigenous aerenchyma formation in lines 96–100

Line 107-108: I think this section requires more explanation with proper references regarding the use of specific agar medium with iron sulfide.

We appreciate the concern raised by the reviewer. We would like to specify that details about the medium and application of FeS are provided in the materials and methods section (lines 450–468), with an appropriate reference given line 450.

Line 121-129: The criteria used to divide the mapping population into two groups (P1-like and P2-like) for the three traits (ROL, root area and root length) are unclear. All three traits are quantitative with continuous variation and I don’t think it is proper to classify the mapping population simply into two groups using arbitrary criteria (there is a similar concern regarding line 262).

In addition, Figure 1 and 2 seem to be redundant presenting the same data in two different formats and I think one of them can be omitted from the main manuscript.

For Fig 1A, 1B, and 1C, axis information is labeled incorrectly – x axis, not y, should be labeled with each trait.

We appreciate the concern raised by the reviewer. We would like to indicate that the grouping of mapping population was not done randomly. Using an R script specific for generating clusters, the mapping population was clustered based on their ROL response (which is the major target trait) compared to that of the parental lines. On this basis, we could identify DH lines with low and high ROL values. We have specified the algorithm in the materials and methods section (lines 486–496)

We appreciate the concern expressed by the reviewer. However, we would like to specify that that from our understanding a normal distribution plot and a box plot are complementary. In some cases the box plot alone may not tends to show symmetric data but may not necessarily mean that the data is normally distributed. This is the reason why we showed the normal distribution and the box plot to clarify this point.

We apologize for the inconvenience. We corrected the labelling in all panels of Figure 1 as suggested (Line 148)

Line 141-151: What is the point of conducing PCA and clustering analysis with only three traits and describing trait correlations using the PCA result? I would suggest conducting simple correlation analysis instead and describing trait correlations directly using actual correlation coefficients.

We appreciate the concern raised by the reviewer. According to the literature, our sample size of 117 lines and 2 parental lines with three variables are eligible to performing a PCA. In this manner, we could visualize how the DH lines for each trait studied are related to each as well as the relationship between traits.

Line 159: Please mention specifically how many of each marker (KASP and Fluidigm) were used for genotyping. What was the total linkage map length in cM?

We specified the number of KASP and Fluidigm markers used in the study in lines 190 and 498. The total linkage map length has been added in Table S1, and a brief reference in lines 199–201.

Line 162-167: These sentences should be rephrased. The flanking markers and additive effects are described for only one QTL while dealing with two separate QTLs.

Please also add more information regarding the QTL for root length along with the LOD threshold used for declaring significant QTLs.

We are thankful to the reviewer for his valuable suggestion. We have rephrased the description of the results in lines 195–196)

We appreciate the suggestion made by the reviewer. As indicated in Table 1, we did not detect any QTL associated with root length.

Figure 4: The legend should provide information on different colors used in the LOD graph.

We added a description of LOD peaks and QTL lines in both panels of Figure 4 (lines 211–214)

Section 3.2: Be consistent with the QTL nomenclature. qROL-2 or qROL-2-1?

We apologize for the inconvenience. We have harmonized throughout the manuscript the indication of qROL-2-1 as suggested.

Line 354-360: Most information described here is redundant with line 333-338.

In Section 4.1, more information is required regarding experimental design. How many seeds were sown per each line and each culture plate? How many biological replications did you use per line?

Is it correct that the height of the culture plate is only 2 cm while the root length measured was in the range of 3 – 9 cm?

We could not see redundancy in the section pointed by the reviewer. In section 4.1 (M&M), we describe the plant materials used and the sterilization process. Nonetheless, we have slightly modified the connecting statement in line 446.

We have specified the number of seeds per plate and their replications in lines 448–447. In essence, two seeds per square plate in triplicate were sown on FeS medium. During our initial trial, three seeds in a square plate resulted in intercrossed roots, which did not help in evaluating ROL of each seedling.

We thank the reviewer for the concern raised. As the reviewer may be aware, to allow the development and growth of roots downward (geotropism) and that of the shoot upwards, the plate we put in more or less 60 degrees inclination and exposed to the light. Therefore, the length of the roots followed the length of the plate not its height (on agar medium).

Section 4.2: Information on root length measurement should be also included. Was it measured directly, or was it also image-based data?

We added a brief description of root length measurement in lines 481–485. Only ROL and RA were image-based but root length was actual recorded measurement with a ruler on a

Section 4.3: More information is required on linkage map construction and QTL analysis. What software was used? What were the functions used for linkage map and QTL? What is the LOD threshold used?

We added a description of the linkage map analysis in lines 505–515.

  1. Nikkhah, A.J.J.R.R. Rice for Ruminants: Race for a Science Under Shadow. 2015, 3, 134.
  2. Asyifah, M.; Abd-Aziz, S.; Phang, L.; Azlian, M.J.J.o.A.P.R. Brown rice as a potential feedstuff for poultry. 2012, 21, 103-110.
  3. Bodie, A.R.; Micciche, A.C.; Atungulu, G.G.; Rothrock Jr, M.J.; Ricke, S.C.J.F.i.S.F.S. Current trends of rice milling byproducts for agricultural applications and alternative food production systems. 2019, 3, 47.

Reviewer 3 Report

In this manuscript, authors did Novel QTL Associated with Aerenchyma-Mediated Radial Oxygen Loss (ROL) in Rice (Oryza sativa L.) under Iron (II) Sulfide. In this study, authors evaluated the root growth habit and ROL rate of a doubled haploid (DH) population (n=117) and the parental lines 93-11 (P1, indica) and Milyang352 (P2, japonica) in response to iron (II) sulfide (FeS) treatment. In addition, we performed a linkage mapping and quantitative trait locus (QTL) analysis on the same population for the target traits. The phenotypic evaluation results revealed that parental lines had distinctive quantitative root growth patterns and ROL percentage, with 93-11 (indica) and Milyang352 (japonica) showing low and high ROL levels, respectively. This was also reflected in their derived population indicating that 93.2% of the DH lines exhibited a high ROL (P2-like) and about 6.8% had a low ROL pattern (P1-like). Furthermore, the QTL and linkage map analysis detected two QTLs associated with the control of ROL from aerenchyma and root area on chromosomes 2 (qROL-2-1, 127 cM, the logarithm of the odds (LOD) 3.04, phenotypic variation explained (PVE) 11.61%) and 8 (qRA-8-1, 97 cM, LOD 4.394, PVE 15.95%), respectively. The positive additive effect (2.532) of qROL-2 indicates that the allele from 93-11 contributed to the observed phenotypic variation for radial oxygen loss. The breakthrough is that the qROL-2 harbors genes proposed to be involved in stress signaling, defense response mechanisms, and transcriptional regulation, among others. Therefore, this study suggests that transcription factor (TF) family members, including TCP7, bHLH, MYB21, PLIM2a, coupled with the stress-responsive genes, such as DEF7, ARF8, EXPA, NIP2, cb5, and LRR based on their annotated functions and previous reports, may play a role in the regulation of ROL level in rice. The overall manuscript is written well. However, for the betterment of this study, I have a few constructive suggestions for the authors.

  1. Please provide morphological evidence also for FeSO4 Stress in rice varieties.
  2. Plot Genome-wide Manhattan plots for grain traits of a DH population.
  3. Do The density map of the pairwise kinship matrix.
  4. Make one hypothetical figure depicting the finding of this study.

Author Response

Title: Novel QTL Associated with Aerenchyma-Mediated Radial Oxygen Loss (ROL) in Rice (Oryza sativa L.) under Iron (II) Sulfide

Manuscript ID plants-1539996

Journal Plants

Section Plant Molecular Biology

Special Issue Development of System to Conventional and QTL Breeding for Rice Cultivars

Point by point reply to reviewers’ comments

We are thankful to the editorial team and anonymous reviewers for their time dedicated to this manuscript. We sincerely appreciate the comments, which helped us improved significantly the content and the quality of the manuscript. We would like to specify that all changes in the manuscript in the first round of revision were highlighted green. We hope that the manuscript in the present form is suitable for publication in the journal.

Reviewer 3

In this manuscript, authors did Novel QTL Associated with Aerenchyma-Mediated Radial Oxygen Loss (ROL) in Rice (Oryza sativa L.) under Iron (II) Sulfide. In this study, authors evaluated the root growth habit and ROL rate of a doubled haploid (DH) population (n=117) and the parental lines 93-11 (P1, indica) and Milyang352 (P2, japonica) in response to iron (II) sulfide (FeS) treatment. In addition, we performed a linkage mapping and quantitative trait locus (QTL) analysis on the same population for the target traits. The phenotypic evaluation results revealed that parental lines had distinctive quantitative root growth patterns and ROL percentage, with 93-11 (indica) and Milyang352 (japonica) showing low and high ROL levels, respectively. This was also reflected in their derived population indicating that 93.2% of the DH lines exhibited a high ROL (P2-like) and about 6.8% had a low ROL pattern (P1-like). Furthermore, the QTL and linkage map analysis detected two QTLs associated with the control of ROL from aerenchyma and root area on chromosomes 2 (qROL-2-1, 127 cM, the logarithm of the odds (LOD) 3.04, phenotypic variation explained (PVE) 11.61%) and 8 (qRA-8-1, 97 cM, LOD 4.394, PVE 15.95%), respectively. The positive additive effect (2.532) of qROL-2 indicates that the allele from 93-11 contributed to the observed phenotypic variation for radial oxygen loss. The breakthrough is that the qROL-2 harbors genes proposed to be involved in stress signaling, defense response mechanisms, and transcriptional regulation, among others. Therefore, this study suggests that transcription factor (TF) family members, including TCP7, bHLH, MYB21, PLIM2a, coupled with the stress-responsive genes, such as DEF7, ARF8, EXPA, NIP2, cb5, and LRR based on their annotated functions and previous reports, may play a role in the regulation of ROL level in rice. The overall manuscript is written well. However, for the betterment of this study, I have a few constructive suggestions for the authors.

We are thankful to the reviewer for the valuable and constructive comments and suggestions made in order to improve the quality and the contribution of our study. We are made the necessary changes accordingly throughout the manuscript, and new figures (data) have been added to the manuscript as suggested.

  1. Please provide morphological evidence also for FeSO4 Stress in rice varieties.

We appreciate the concern raised by the reviewer. We have added the phenotype of parental lines and two DH lines with the highest and lowest ROL response as Figures 2D,E

2.       Plot Genome-wide Manhattan plots for grain traits of a DH population.

We are thankful to the review for the suggestion made, and strongly appreciate it. However, we would like to specify that several strategies for the detection of novel QTLs have been reported, including linkage mapping and QTL analysis or Genome-wide association study (GWAS). In this study, we have used a linkage mapping and QTL analysis strategy with regard to the number and the density of the KASP and Fluidigm markers used. The use of GWAS to generate a Manhattan plot is appropriate where more markers are available with a high density. Therefore, we did not find it appropriate to use a combine GWAS-linkage mapping strategy to perform the experiments.

3.       Do the density map of the pairwise kinship matrix.

We are thankful to the reviewer for the valuable suggestion. We have included the density map of KASP and Fluidigm makers across the rice genome as Figure S1.

4.       Make one hypothetical figure depicting the finding of this study.

We strongly appreciate the reviewer’s suggestion to include a sort of model summarizing the findings of our study. We have constructed a hypothetical signaling model (Figure 6) using the phenotypic data (Figure 2) and the recorded transcript accumulation results (See Figure 5), coupled with the existing literature in the field.

Round 2

Reviewer 2 Report

I think the academic value of the manuscript has been significantly improved after the extensive revision the authors carried out, especially by adding more results from the gene expression experiments. I have few more comments as below.

- Line 152-159: I still think that this part of the manuscript does not make sense. The authors’ explanations on grouping the mapping population for ROL, root area and root length into two groups are not convincing. The authors used the threshold values of 16.4% (ROL), 12.7% (root area), and 6.8% (root length) for dividing the mapping population into two groups for each trait. Were these values determined based on the clustering analyses the authors claimed as in lines 494-495 and the cover letter? 12.7% for root area and 6.8% for root length are simply the values of one parent (93-11). What is the basis of using 16.4% to divide the population into high ROL and low ROL groups? This grouping puts 6.8% of the mapping population at low ROL group as described in line 153, which is not consistent with Fig. 3A where nearly 50% of the population is depicted as low ROL lines.   

- The abstract should be shortened to meet the journal’s guideline (~200 words maximum).

Author Response

Title: Novel QTL Associated with Aerenchyma-Mediated Radial Oxygen Loss (ROL) in Rice (Oryza sativa L.) under Iron (II) Sulfide

Manuscript ID plants-1539996

Journal Plants

Section Plant Molecular Biology

Special Issue Development of System to Conventional and QTL Breeding for Rice Cultivars

Point by point reply to reviewers’ comments

We are thankful to the editorial team and anonymous reviewer for his time dedicated to this manuscript. We sincerely appreciate the comments, which helped us improved significantly the content and the quality of the manuscript. We would like to specify that all changes in the manuscript in this second round of revision are highlighted yellow. Previous changes remained highlighted green. We hope that the manuscript in the present form is suitable for publication in the journal.

Reviewer 2

I think the academic value of the manuscript has been significantly improved after the extensive revision the authors carried out, especially by adding more results from the gene expression experiments. I have few more comments as below.

We are thankful to the reviewer for his valuable comments that have helped us improve the scientific value of our manuscript.

- Line 152-159: I still think that this part of the manuscript does not make sense. The authors’ explanations on grouping the mapping population for ROL, root area and root length into two groups are not convincing. The authors used the threshold values of 16.4% (ROL), 12.7% (root area), and 6.8% (root length) for dividing the mapping population into two groups for each trait. Were these values determined based on the clustering analyses the authors claimed as in lines 494-495 and the cover letter? 12.7% for root area and 6.8% for root length are simply the values of one parent (93-11). What is the basis of using 16.4% to divide the population into high ROL and low ROL groups? This grouping puts 6.8% of the mapping population at low ROL group as described in line 153, which is not consistent with Fig. 3A where nearly 50% of the population is depicted as low ROL lines.

We would like to thank the reviewer for his concern. In Figure 3A, we displayed a cluster of the mapping population only based on their recorded ROL (as the target trait) rates relative to the parental lines. We used the parents ROL, RA, and RL phenotypic responses as reference to compare the changes in the derived population. Then, we used the specific R fuction indicated lines 494-495 to cluster them, knowing that parental lines recorded distinctive phenotypic responses, one showing a high and the other showing a low ROL rates. Without a reference threshold, clustering would be a random exercise.

In the efforts to clarify the logic in this section, we have provided a briefly description in lines 147-160. We have put the parental description in the first position followed by the description of the mapping population. The clusters of the mapping population is now shown separately by trait (Figures 3A–C) (lines 179–182).

- The abstract should be shortened to meet the journal’s guideline (~200 words maximum).

The concern raised by the reviewer is highly appreciated. We have tried to reduce the word count in the abstract as much as we could. However, we could not go down to 200 words to avoid emptying the abstract from its substance. We hope that this could be understood by the reviewer.

Reviewer 3 Report

I am happy with the author's comments and the manuscript can be accepted in its current format.

Author Response

We are thankful to the reviewer for his valuable and constructive comments, which have helped us improve the quality of our manuscript.